# Fault Diagnosis for Aircraft Hydraulic Systems via One-Dimensional Multichannel Convolution Neural Network

**Kenan Shen [1,*]** and **Dongbiao Zhao [2]**

1   College of Automation Engineering, Nanjing University of Aeronautics and Astronautics, Nanjing 211106, China
2   College of Mechanical and Electrical Engineering, Nanjing University of Aeronautics and Astronautics, Nanjing 210016, China; zdbme@nuaa.edu.cn
*   Correspondence: kekedear@nuaa.edu.cn

**Abstract:** Detecting the faults in hydraulic systems in advance is difficult owing to the complexity associated with such systems. Hence, it is necessary to investigate the different fault modes and analyze the system reliability in order to establish a method for improving the reliability and security of hydraulic systems. To this end, this paper proposes a novel one-dimensional multichannel convolution neural network (1DMCCNN) for diagnosing fault modes. In this work, a landing gear hydraulic system was constructed with a normal model and a fault model; five types of faults were considered. Pressure signals were extracted from this hydraulic system, and the extracted signals were subsequently input into the convolution neural network (CNN) as multichannel data. Thereafter, the data were subjected to a one-dimensional convolution filter. The differences between channels were used to enhance features. The features obtained in this manner were compared for fault diagnoses. Furthermore, this proposed method was verified via simulations; the simulation results indicated that the precision of the 1DMCCNN was considerably higher than that of conventional machine learning algorithms.

**Keywords:** aircraft hydraulic system; convolution neural network; one-dimensional multichannel convolution neural network algorithm; fault diagnosis

## 1. Introduction

The hydraulic system of the landing gear is one of the most important components of the hydraulic systems in aircraft. Notably, the reliability of the hydraulic system has a significant influence over the safety of an aircraft. As reported previously, approximately 30% of all mechanical failures in an aircraft stem from issues in the hydraulic system. Accordingly, maintenance operations related to hydraulic systems account for one-third of the entire mechanical maintenance work [1]. However, identifying the causes of failures in hydraulic systems remains difficult [2]. Unlike other systems, all the components and the oil in hydraulic systems operate within a closed circuit; hence, fault diagnoses for such systems are not intuitive.

In an aircraft, hydraulic control systems are often accompanied by backup systems; to guarantee the safety of an aircraft, ensuring the reliability of these systems is essential. In this regard, monitoring the condition of hydraulic systems is significantly important in both academic and industrial fields [3–5]. Evidently, to ensure the safe and reliable operation of aircrafts' hydraulic system, fault diagnoses of the hydraulic system are crucial [6–8]. Hence, it is desirable to develop fault diagnosis schemes that can identify abnormal conditions in hydraulic systems.

Fault diagnoses for aircrafts' hydraulic systems involve four stepped stages: subjective diagnoses by technicians, measurement-based diagnoses using simple instruments, diagnoses using model analysis, and intelligent fault diagnoses [9–11]. Currently, measurements and diagnoses using simple instruments are mainly used for aircrafts' hydraulic

systems. With regard to intelligent fault diagnoses, the commonly used approaches entail analyzing the pressure signals of aircrafts' hydraulic systems via wavelet packet decomposition or the information entropy method, followed by extracting important features into neural networks or support vector machines (SVM) models for fault diagnoses and classification [12–14].

Thus far, several studies have been devoted toward fault diagnoses in hydraulic systems. For instance, Dao et al. [15] designed an active fault-tolerant control (FTC) system for an *n*-degree-of-freedom (*n*-DOF) hydraulic manipulator with internal leakage faults and mismatched matched lumped disturbances; this active FTC system achieved suitable position tracking performance under conditions involving single faults and multiple simultaneous faults. Jin et al. [16] reported that the piston seal wear in hydraulic cylinders is a primary factor resulting in internal leakages; they applied a wavelet transform as a feature extractor to transform raw oil pressure data into feature vectors comprising the wavelet packet energy, energy entropy, energy variance, and the root mean square of the wavelet detailed coefficient. They proposed a fault detection and identification scheme that was, based on leakage experiments and simulation data, capable of effectively detecting faults. Maddahi et al. [17] introduced a multiscale analysis, which included measures for the correlation entropy and wavelet detail coefficients, for the detection of internal leakages in electro-hydrostatic actuators. Many types of faults may occur in hydraulic systems; these include actuator faults caused by internal leakages, external leakages, and a reduction in the supply pressure and sensor faults caused by issues in the pressure and position sensors [18,19]. The aforementioned methods achieved suitable results; however, considering faults at the system level, these previous methods may not afford good results owing to the difficulties associated with preprocessing and feature extraction at the system level. Therefore, other scholars and researchers have studied the fault diagnosis of an aircraft's hydraulic system [20–23].

To address this problem, this paper proposes a novel method for diagnosing faults in aircrafts' hydraulic systems, based on a one-dimensional multichannel convolution neural network (1DMCCNN). In the proposed method, pressure signals of the hydraulic system are extracted via simple normalization and directly input into the CNN. Notably, this approach does not entail complex data preprocessing and feature extraction to ascertain the end-to-end fault-diagnosis mode. Owing to the structure of the CNN, the pressure signals acquired from multiple sensors located at different positions in the aircraft's hydraulic system can be directly input into the network as multichannel input signals to achieve multisensor fusion; furthermore, the differences between the sensor results can be used to enhance features. More importantly, simulation results reveal that the precision of the proposed method is significantly better than that of conventional methods.

The remainder of this paper is organized as follows. Section 2 presents the improved 1DMCCNN algorithm for the fault diagnosis of hydraulic systems. Section 3 describes the hydraulic systems built using AMESIM, including the normal and fault modes; the simulation results used to verify the proposed 1DMCCNN method are also presented. Section 4 explains the analyses of the simulation results and also presents a comparison between the 1DMCCNN method and other conventional machine learning algorithms; as discussed in this section, the proposed algorithm shows good performance in terms of the training time, model size, and other aspects. Lastly, the conclusions of this work and the scope of future research are outlined in Section 5.

## 2. Fault Diagnoses Based on Improved 1DMCCNN Algorithm

LeNet5, the first CNN model, was proposed by Le et al. [24,25]. This network comprised classical CNN structures such as the convolution layer, pooling layer, and full connection (FC) layer. Thereafter, Krizevsky et al. introduced Alex Net [26] in 2012; since winning the image classification contest involving ImageNet, a large image database, deep learning models based on the CNN structure have undergone rapid developments [27–29]. CNNs do not require complex and tedious data preprocessing and feature extraction pro-

cesses; consequently, they are significantly advantageous for high-dimensional data such as images and long time-series signals.

### 2.1. One-Dimensional Convolution

Unlike ordinary neural networks, CNNs feature a convolution layer structure with weight sharing [24]. Let $H_l$ be the input feature graph of the convolution layer; in this case, the original feature graph is $H_0$, $S$ is the result of the convolution operation, $H_{l+1}$ is the output feature graph of the convolution layer, $K$ is the convolution core, $i$ is the index of the feature graph, $p$ is the index on the convolution core, $b$ is the offset, and $f(x)$ is the activation function. Thus, the one-dimensional convolution operation can be expressed as follows:

$$S(i) = \sum_p H_l(i + p)K(p) \tag{1}$$

$$H_{l+1} = f(S + b) \tag{2}$$

### 2.2. One-Dimensional Pooling

Pooling, also known as downsampling [30], is typically employed after the convolution layer to reduce dimensionality and extract effective features. Pooling operations can be divided into maximum pooling and mean pooling. Let layer $l$ be the pooling layer, $k$ be the size of the pooling window, and $p$ be the index on the pooling window. Thus, the operation formulas for maximum pooling and mean pooling can be expressed as in Equations (3) and (4), respectively:

$$H_{l+1}(i) = \max_{p \in K} H_l(i + p) \tag{3}$$

$$H_{l+1}(i) = \frac{1}{k}\sum_p H_l(i + p) \tag{4}$$

### 2.3. Fault Diagnosis with CNN

CNN-based fault diagnosis is a data-driven, supervised learning classification problem. First, data pertaining to different states of the system are collected through experiments or simulations; these data are then classified and normalized. Subsequently, the data are resampled, that is, multiple subsets with the same length (samples) are intercepted using a sliding window of a long time-series signal; as the subsets can be repeated, resampling can help enhance the data set. These samples are divided into the training and test sets. The training set is used to train the CNN model and help the network learn the fault classification information contained in the data, whereas the test set is used to verify the generalization ability of the trained network. Fault diagnosis results can then be obtained by inputting the signals to be diagnosed into the trained network. The corresponding flowchart for a CNN-based fault diagnosis is depicted in Figure 1.

In this network, one-dimensional convolution is used to process one-dimensional time-series signals, while the multisensor fusion is realized via multichannel convolution.

### 2.4. Multisensor Fusion

CNNs can directly receive gray images from multiple channels during training. Thus, the pressure signals collected by sensors at different locations can be directly input into the CNN through different channels simultaneously, to achieve multisensor fusion [26]. Let $c$ be the index on the channel scale and $H_l(i + p, c)$ be the feature graph corresponding to $c$; then, the multichannel, one-dimensional convolution formula can be expressed as follows:

$$S(i) = \sum_c \sum_p H_l(i + p, c)K(p) \tag{5}$$

Considering the common fault forms in aircrafts' hydraulic systems, the signals from three sensors—pump outlet pressure sensor, oil filter outlet pressure sensor, and actuator pressure sensor—can be employed as the three-channel input for the CNN (hereinafter, this

model is denoted as 1DMCCNN-v1). Further analyses reveal that the difference between the pressure values provided by the pump outlet pressure sensor and the oil filter outlet pressure sensor can offer additional information, as shown in Figure 2.

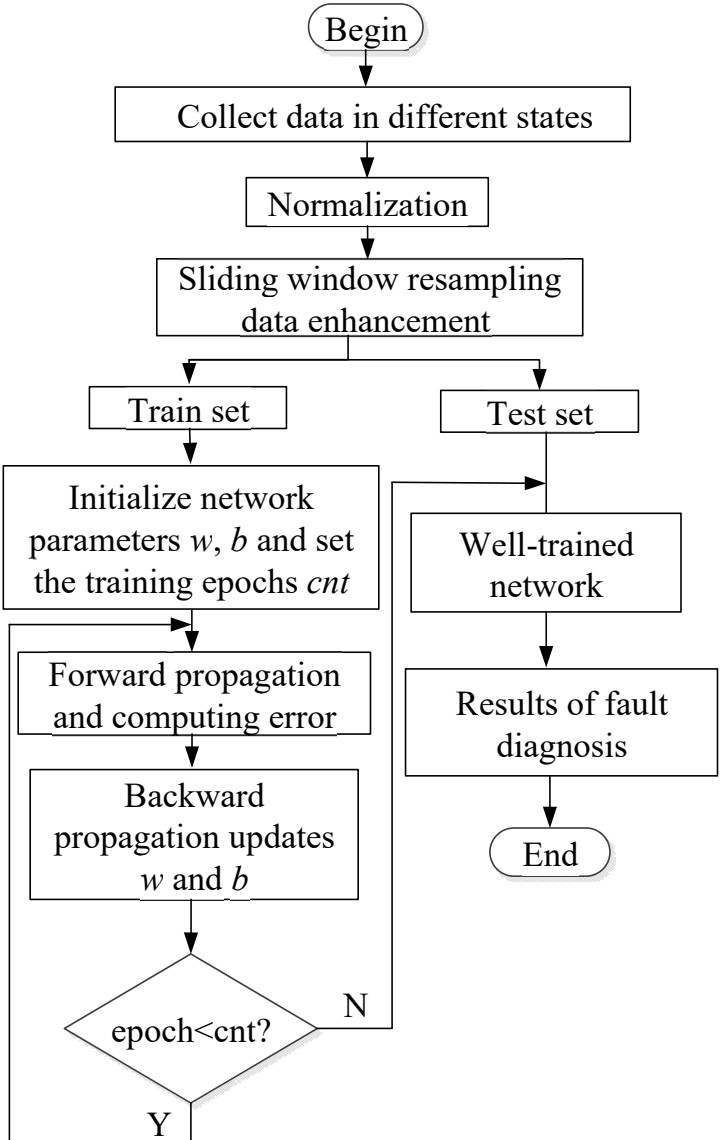

**Figure 1.** Flowchart for CNN-based fault diagnoses.

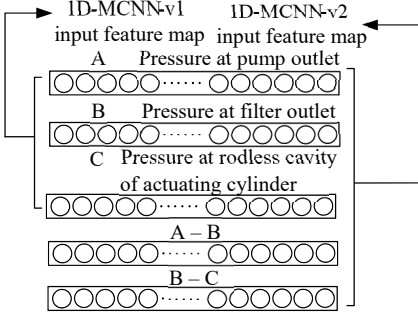

**Figure 2.** 1DMCCNN-v1 and 1DMCCNN-v2.

Similarly, the differences between the pressure values provided by the oil filter outlet pressure sensor and the rodless chamber pressure sensor of the actuator barrel can be

made, so that the information of five channels can be obtained (hereinafter denoted as 1DMCCNN-v2).

### 2.5. Structural Design and Improvement of 1DMCCNN

Two pairs of convolution pooling layers were connected with two FC layers and one softmax layer, as shown in Figure 3. The convolution layer was responsible for feature extraction, whereas the pooling layer (where maximum pooling was used) was responsible for dimensionality reduction. The softmax layer and the FC layer classified the feature maps extracted via convolution pooling. Typically, the size and step size of the convolution pooling are initially determined based on experience; thereafter, optimal values were determined using the parameters. The input layer was linked to a fully connected layer having 1800 nodes. The output of the fully connected layer was transformed into 6 classes of faults by the reshape function. The fully connected layer was followed by 2 consecutive two-dimensional deconvolutional layers and 2 poolings. The parameters of each layer are presented in the Figure 3. For 1DMCCNN-v2, it was necessary to ensure a difference between the input features such that the input feature size could be changed to $1 \times 600 \times 5$, while the other parameters remained unchanged.

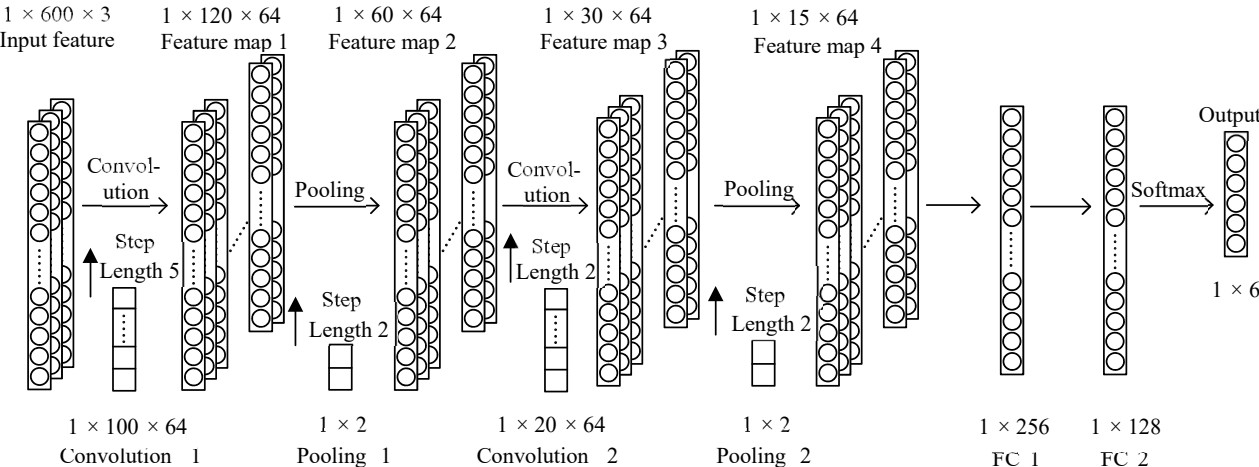

**Figure 3.** Structure of 1DMCCNN.

### 3. Simulations

The simulation model for the LG hydraulic system was built using LMS AMESIM, AMESIM is a modular modeling and simulation software. Considering the aircraft's landing gear's (LG) hydraulic system as the research object, during operation, high-pressure oil from the oil supply system enters the lock cylinder, hydraulic lock, and actuating cylinder to facilitate the retraction and lowering of the LG. Using AMESIM, a simulation model of the working loop of the LG's hydraulic cylinder was built, based on the working principle of the hydraulic cylinder and the hydraulic library and design library for hydraulic components, as shown in Figure 4. In Figure 4, there are 3 subsystems which are oil supply system, servo-actuation system and power transfer unit(PTU). The oil supply system can be seperated to three part, system 1 and system 2 are working system, and system 3 is emergency system, it will not work, untial system 1 and system 2 break down. Servo-actuation system includ the landing gear system, and other hydraulic working systems which are not considered in this paper. The four dotted lines(red) in Figure 4 means the servo-actuation system and the three oil supply systems.

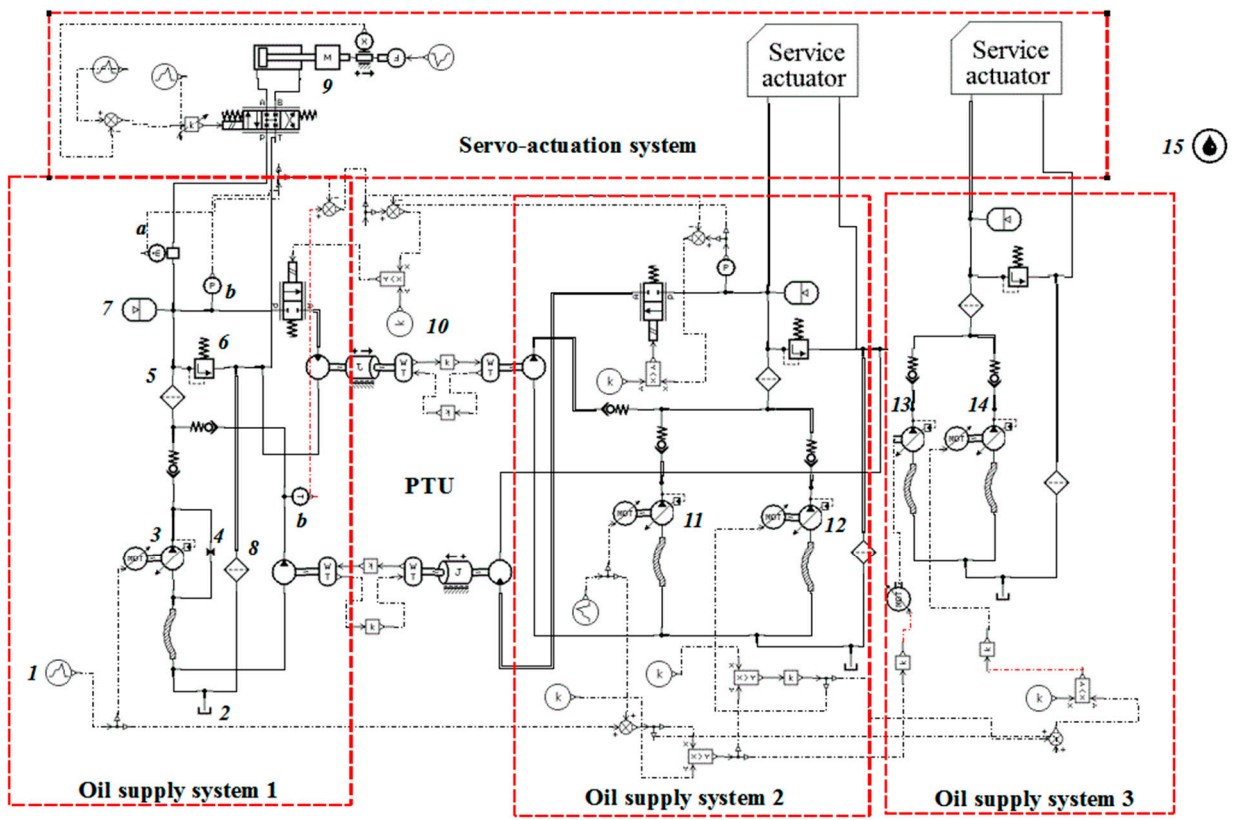

**Figure 4.** Landing gear hydraulic system with three oil supply systems, built using AMESIM.

### 3.1. Normal Model

In this work, the hydraulic system of the A320 large passenger aircraft was adopted as the simulation object, and LMS AMESIM was employed to build the hydraulic system model, as shown in Figure 2. The hydraulic system of the A320 passenger aircraft is composed of three relatively independent systems: the green system (system 1), blue system (system 3), and yellow system (system 2). The green system is powered by the left engine drive pump (EDP); the blue system is powered by the electric pump (EMP) under normal conditions and by the ram air turbine (RAT) in the case of emergencies. Furthermore, the yellow system is powered by the right EDP and an electric pump. The main parameters used in the simulation are listed in Table 1. The number column in Table 1 represents the hydraulic components in Figure 4. A linear model was selected for the simulations; all nonlinear problems were neglected. The normal simulation model was built considering stable conditions for the system. Under these normal operating conditions for the system, supply system 1 (Figure 2) and actuator 9 (Figure 2) were employed for the normal-mode LG's hydraulic system in the simulations and analyses.

**Table 1.** Simulation parameters for the A320 hydraulic system.

| Number | AMESIM Element | Key Parameter | Value | Meaning |
|---|---|---|---|---|
| 1 | signal03 | Output | 5000 | Set the shaft speed to 5000 r/min, that is, the zero-flow pressure of the pump is 3000 psi. |
| 3 | accumulator_2 | Gas precharge pressure (psi)<br>Accumulator volume (L) | 1885<br>2.62 | Accumulator reduces pressure pulses, as an emergency pressure source. |
| 4 | presscontol01 | Relief valve cracking pressure (psi) | 3436 | Pressure relief valve for system discharge. |

**Table 1.** *Cont.*

| Number | AMESIM Element | Key Parameter | Value | Meaning |
|---|---|---|---|---|
| 5 | tank01 | Tank pressure (psi) | 50 | Booster tank, preboost to 50 psi. |
| 7 | pump13 | Nominal shaft speed (r·min$^{-1}$) | 5000 | Left engine drive pump (EDP). |
| 11–14 | pump13 | Nominal shaft speed (r·min$^{-1}$) | 5000 4166 | Right EDP, yellow system EMP, blue system EMP, RAT. Rated pressure of RAT is 2500 psi. |
| 10 | constant_3 | Constant value | 34.4738 | PTU opens when the pressure difference between green and yellow systems is 34.4738 bar. |

### 3.2. Fault Mode

The fault model was built according to the normal model, while considering five different common fault modes: pump leakage (PL), filter block (FB), actuator inner leakage (AL), servo valve block (SVB), and oil pollution (OP). The corresponding fault-mode parameters are listed in Table 2. The specific parameter settings of the components in this table were referred from a previous study [31]. Thus, the common faults of hydraulic systems can be simulated by varying the parameters of the components.

**Table 2.** Fault simulation in the A320 hydraulic system.

| Number | Fault Category and Category Number | Key Parameter | Normal Value | Fault Value |
|---|---|---|---|---|
| 6 | Pump leakage—1 | Equivalent orifice diameter (mm) | 0.1–0.3 | 1–2 |
| 2 | Filter blockage—2 | Equivalent orifice diameter (mm) | 5–7 | 3–4 |
| 9 | Actuating Cylinder inner leakage—3 | Leakage coefficient (L·min$^{-1}$·bar$^{-1}$) | 0–0.01 | 0.03–0.05 |
| 10 | Servo valve blockage—4 | Equivalent orifice diameter (mm) | 5–7 | 3–4 |
| 8 | Oil pollution—5 | Air content (%) | 0.1–0.3 | 5–15 |

This simulation model was built according to the pump leak model, considering the oil leakage coefficient. To further understand the impact of the support fault mode on system stability, different values were adopted, while keeping the other parameters constant. Notably, pump leakage was selected as the fault mode, the normal value was approximately 0.1 mm, and the failure value was 0.8 mm.

### 3.3. Simulation Results

Using oil supply system 1 (Figure 4) as the simulation system, a single actuator (element 9 in Figure 4) was simulated and analyzed. The simulation time was 18 s, with the actuator closed for 0–2 s, opened for 2–8 s, closed for 8–10 s, opened for 10–16 s, and closed for 16–18 s, after which the simulation was halted. Pressure sensors were installed at the pump outlet (Figure 5(a)), oil filter outlet (Figure 5(b)), and rodless chamber of the actuator (Figure 5(c)). Considering pump leakage as an example, the pressure signal curves obtained from the three sensors under the fault and normal conditions are depicted in Figure 5.

Furthermore, considering oil filter blockage as an example, Figure 6 shows the curve representing the differences in the pressures at the oil filter outlet and pump outlet under normal and fault conditions.

More evident differences were noted between the normal signals and the fault signals in the new feature map after making a difference. Hence, adding the original feature map to increase the number of effective features is more conducive to fault diagnoses and classification.

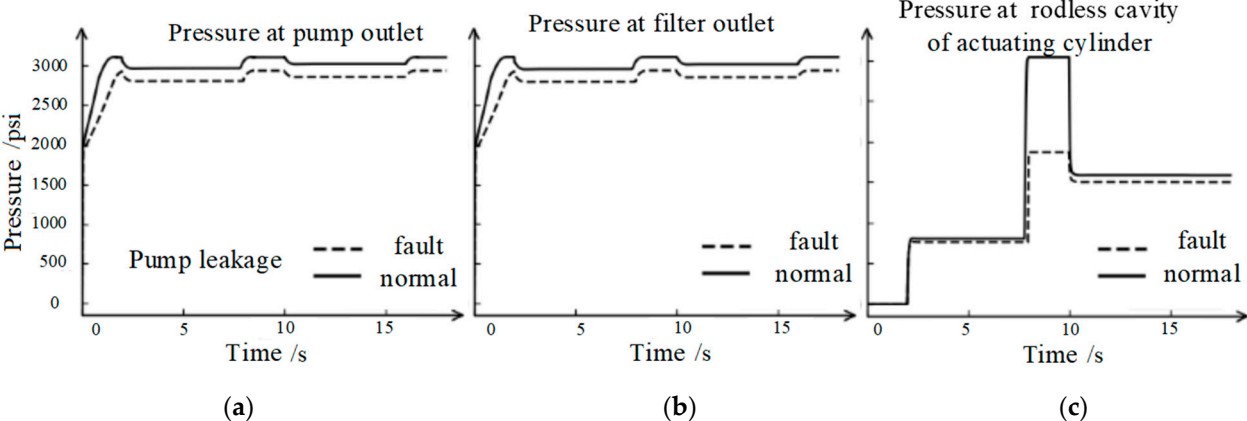

**Figure 5.** Variation in pressure signals under normal and fault conditions.

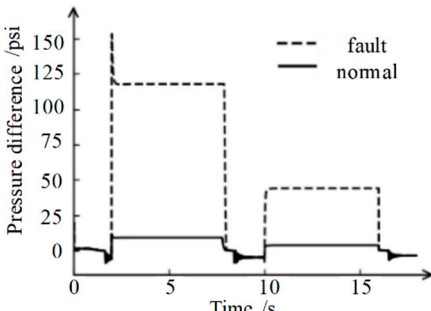

**Figure 6.** Variation in pressure difference under normal and fault conditions.

## 4. Analyses and Comparison

### 4.1. Data Collection and Processing

For a sampling period of 0.01 s, 1800 data points can be collected within 18 s for a curve; in this manner, three curves can be obtained using three sensors located at different positions. For multiple simulations, normal and fault classes can be used to enhance datasets by fine-tuning the parameters of the appropriate-range classes. These data were processed according to the flow chart shown in Figure 4. Each sample contained three curves (i.e., three sensors) with a length of 6 s. To simulate the collected signals under actual situations, original signals were added to the dataset after including a certain amount of uniform-distribution noise and normal-distribution noise. In this study, 80% of all the samples were used as the training set for the CNN, and the remaining 20% were used as the test set to verify the generalization effect of the network.

### 4.2. Analysis of Fault Diagnosis Results

The confusion matrices of 1DMCCNN-v1 and 1DMCCNN-v2 for the test set are shown in Tables 3 and 4, respectively. The lower right corner of the table represents the total accuracy. Owing to the difference between the feature diagrams, under certain fault categories, the precision and recall of the v2 network are markedly higher than those of the v1 network; the overall precision is also improved.

Table 5 lists the optimal values for the network parameters, where lr is the learning rate, and batch size refers to the number of batch samples. The dropout rate refers to the random resetting of the neuron output to zero with a certain probability during training, which can help reduce overfitting. The optimal parameter was obtained by an actual simulation of the aircraft's hydraulic system and multiple experiments.

**Table 3.** Confusion matrix for 1DMCCNN-v1.

| Prediction Real | 0 | 1 | 2 | 3 | 4 | 5 | Recall /% |
|---|---|---|---|---|---|---|---|
| 0 | 351 | 0 | 0 | 1 | 1 | 1 | 99.2 |
| 1 | 0 | 61 | 0 | 0 | 0 | 0 | 100 |
| 2 | 1 | 0 | 75 | 1 | 1 | 0 | 96.2 |
| 3 | 1 | 0 | 0 | 59 | 0 | 1 | 96.7 |
| 4 | 1 | 0 | 1 | 1 | 54 | 0 | 94.7 |
| 5 | 2 | 0 | 0 | 0 | 0 | 77 | 97.5 |
| Precision /% | 98.7 | 100 | 98.7 | 95.2 | 96.4 | 97.5 | 98.2 |

**Table 4.** Confusion matrix for 1DMCCNN-v2.

| Prediction Real | 0 | 1 | 2 | 3 | 4 | 5 | Recall /% |
|---|---|---|---|---|---|---|---|
| 0 | 383 | 0 | 0 | 0 | 1 | 0 | 99.7 |
| 1 | 0 | 61 | 0 | 0 | 0 | 0 | 100 |
| 2 | 0 | 0 | 78 | 0 | 0 | 0 | 100 |
| 3 | 0 | 0 | 0 | 61 | 0 | 0 | 100 |
| 4 | 2 | 0 | 0 | 0 | 55 | 0 | 96.5 |
| 5 | 1 | 0 | 0 | 0 | 0 | 78 | 98.7 |
| Precision /% | 99.2 | 100 | 100 | 100 | 98.2 | 100 | 99.4 |

**Table 5.** Optimal parameter values for 1DMCCNN.

| Parameter Name | Value |
|---|---|
| Convolution kernel size | $(1 \times 100, 1 \times 20)$ |
| Convolution step length | $(5, 2)$ |
| Pooling filter size | $(1 \times 2, 1 \times 2)$ |
| Pooling step length | $(2, 2)$ |
| Initial learning rate (lr) | 0.001 |
| lr decaying | $lr = lr \times 0.9/\text{epoch}$ |
| batch size | 128 |
| FC layer dropout rate | 0.4 |
| Training epochs | 20 |
| Optimizer | AdamOptimizer |

To demonstrate the effect of the multichannel convolution in the multisensor fusion, fault diagnoses were conducted using the pump outlet and oil filter outlet sensors, without the actuator rod cavity. The input layer of the CNN was changed to accept one-channel data, rather than three-channel data; the other parameters remained unchanged. Figure 7 shows the results of the fault diagnosis and classification using different sensor pressure signals. A, B, and C represent the sensors located at different positions.

As is evident from the graph, the precision when using a single sensor is considerably lower than that when using multisensor fusion. Moreover, the errors caused by the sensors located at different positions exhibit a clear tendency. For instance, the precision for leakage in the pump remains at 100% when using the pressure sensor at the pump outlet alone. This is because the pressure signal from the sensor contains sufficient information for diagnosing faults. However, the precision for a servo valve blockage and oil filter blockage are significantly lower; this is because these sensors do not offer sufficient information pertaining to these two faults.

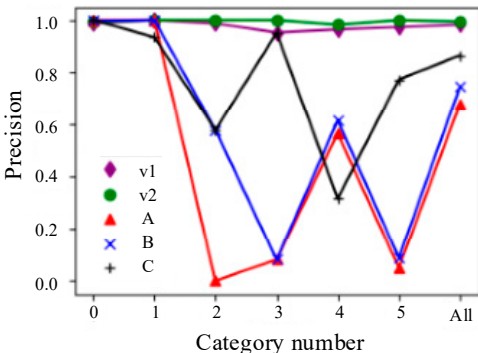

**Figure 7.** Fault diagnosis results for different channel numbers.

### 4.3. Comparison of Proposed Method and Conventional Machine Learning Algorithms

One-dimensional data of $1 \times 1800$ can be obtained via transverse stitching of the pressure signals from three sensors, which can then be input into the backpropagation (BP) neural network and converted into a three-channel 2D map of $20 \times 30 \times 3$; this can be input into the 2DCNN. Figure 8 presents the decline in the loss during the training process for 1DMCCNN-v1, 2DCNN, and the BP neural network. It is evident from the figure that the results of 2DCNN are slightly worse than those of 1DMCCNN-v1; further, the expression ability of the BP neural network is clearly inadequate.

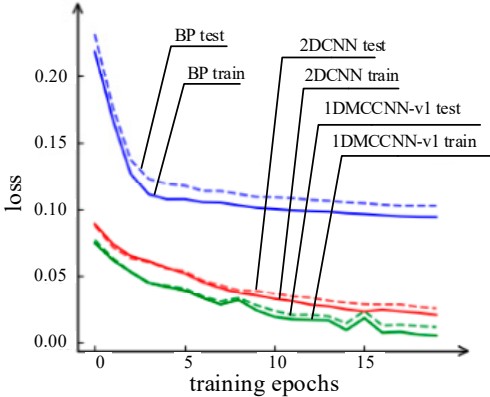

**Figure 8.** Decline in loss during training for proposed and conventional methods.

Table 6 compares the 1DMCCNN-v1, 1DMCCNN-v2 algorithms with several common machine learning algorithms. As can be seen, the test set precision of 1DMCCNN-v2 is the highest; its training speed is only second to that of the BP neural network, with regard to algorithms that need to be trained. Furthermore, the model is smaller than most other models, with the exception of the linear SVM.

**Table 6.** Comparison of common machine learning algorithms.

| Algorithm | Training Precision | Test Precision | Training Time /s | Model Size /MB |
|---|---|---|---|---|
| 1DMCCNN-v1 | 0.990 | 0.982 | 16 | 4.52 |
| 1DMCCNN-v2 | 0.998 | 0.994 | 16 | 4.66 |
| 2DCNN | 0.986 | 0.957 | 24 | 15.5 |
| BP | 0.649 | 0.647 | 7 | 6.83 |
| SVM | 0.902 | 0.732 | 46 | 0.084 |
| KNN | 1 | 0.903 | needless | 82.6 |
| LSTM | 0.952 | 0.943 | 36 | 8.46 |

## 5. Conclusions

In this paper, an improved 1DMCCNN-based design method for hydraulic systems was proposed. A new hydraulic system was built using AMESIM; this model was used to simulate the LG's hydraulic system of an aircraft under normal and fault conditions; the fault model of the system involved five types of faults.

In this work, an improved 1DMCCNN method was employed to diagnose hydraulic system faults. Under this approach, pressure signals were extracted from sensors located at different positions in the aircraft's hydraulic system; these signals were normalized as multichannel data and input into the 1DMCCNN via one-dimensional convolution operations. The simulation results proved the effectiveness of the improved 1DMCCNN method.

Notably, differences between the sensor signals can strengthen effective features and improve the precision of fault diagnoses. Further, a multichannel convolution structure can help realize multisensor fusion, which, in turn, can effectively improve the precision of fault diagnoses, as compared with that when using a single sensor. Moreover, compared with conventional machine learning algorithms, the proposed algorithm achieved the highest accuracy and exhibited good performance in terms of training time, model size, and other aspects.

Although the method in this paper achieved good results, it is worth pointing out that the 1DMCCNN method is a data-driven fault-diagnosis method, which suffers from data imbalance in the process of data acquisition. Moreover, under the actual working status of the aircraft's hydraulic system, the fault data and normal data are seriously imbalanced, and the time under the normal flight status is much greater than that under the fault status. In other words, in most cases, researchers encounter a fault data imbalance, which decreases the accuracy of fault diagnosis. In future work, the authors will pay attention to the data imbalance problem and find a way to provide an effective solution to the data imbalance between the fault data and normal data of the aircraft hydraulic system.

**Author Contributions:** Conceptualization, K.S. and D.Z.; methodology, K.S.; software, K.S.; formal analysis, K.S.; resources, K.S. and D.Z.; writing—original draft preparation, K.S.; writing—review and editing, K.S. and D.Z.; supervision, D.Z.; project administration, D.Z. All authors have read and agreed to the published version of the manuscript.

**Funding:** This research received no external funding.

**Conflicts of Interest:** The authors declare no conflict of interest.

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
