# Peer review of "Fault Diagnosis for Aircraft Hydraulic Systems via One-Dimensional Multichannel Convolution Neural Network"

_actuators, doi:10.3390/act11070182_

Round 1

Reviewer 1 Report

Paper title: Fault Diagnosis for Aircraft Hydraulic Systems via One Dimensional Multichannel Convolution Neural Network

Paper ID: actuators-1782464

The manuscript is of very high quality. It is well organized. The case study is well-conceived. The applied methodology is well described. The authors also gave a comparative analysis of the proposed model and confirmed the quality of the obtained results.

 I suggest that the authors make minor refinements to the paper:

 . Improve literature analysis with 5-10 recent papers (2020-2022).

. Better highlight novelty in the study.

. Show in detail the advantages and limitations of the proposed methodology and this study.

. Add future research.

I wish you all the best with this manuscript and other ones in the future. 

Reviewer 2 Report

This paper proposes a novel one-dimensional multi-channel convolution neural network (1DMCCNN) for diagnosing fault modes. The work sounds interesting and the result seems correct. Before acceptance, some suggestions are given below:

1, In Section 2.5, more explanations or clarifications are suggested to be given to show the improvement of 1DMCCNN.

2,A short explanation is suggested to be given on how to obtain Table 5.

3, The language is suggested to be polished.

4, Some works on data-driven based fault diagnosis are suggested to be updated, e.g., An iterative reassignment based energy-concentrated TFA post-processing tool and application to bearing fault diagnosis, Measurement, 193: 110953, 2022.

Reviewer 3 Report

From my point of view, maybe I missed something:

The description of the methods is quite general.

I do not understand in Fig. 1 the test "epoch < cnt ".

What reduces the number of operations?

Where did the values in Table 2 come from?

Was it just model training or was it tested in practice?

Row 262 - A, B and C are sensors in different positions - where and why were they located like this?

Reviewer 4 Report

Manuscript

Title:Fault Diagnosis for Aircraft Hydraulic Systems via One 2 Dimensional Multichannel Convolution Neural Network”

Authors: Kenan Shen, Dongbiao Zhao

Dear Authors

I revised the manuscript: " Fault Diagnosis for Aircraft Hydraulic Systems via One 2 Dimensional Multichannel Convolution Neural Network" submitted to the “Actuators” Journal. The paper is very interesting. However, I have some concerns, which need to be addressed.

Line 2-3. Article topic

The topic of the article is comprehensible and reflects well the scope of the presented issues.

Abstract:

Line 16. „CNN” Please explain the introduced abbreviations of terms, possibly as soon as they are introduced in the content of the article. A single explanation is sufficient in the execution of the explanation.

CNN - convolution neural network?

The abstract presents the key elements of the article well: research question, methodology, results and conclusions.

Keywords. Keywords are chosen correctly.

One caution, abbreviations and acronyms are difficult for readers to recognise. I suggest to avoid abbreviations and enigmatic symbolism in the presentation of keywords.

1.    Introduction

Line 47. „….SVM….” Please explain the introduced abbreviations of terms, possibly as soon as they are introduced in the content of the article. A single explanation is sufficient in the execution of the explanation.

Line 83. „….AMESIM….” Please explain the introduced abbreviations of terms, possibly as soon as they are introduced in the content of the article. A single explanation is sufficient in the execution of the explanation.

Line 81-89 The scope of the work is clearly defined. There is a lack of a specified research goal that unambiguously formulates the research question. Without this, verification of the logical arrangement of information in the article is difficult. Please indicate clearly the initial purpose of the conducted research.

Line 23-89. The numbering of literature sources in the text is in accordance with formal requirements. The content of the chapter introduces the reader to the problematic issue in a logical and understandable way.

Line 54, 60."....Jin et al....." Please use a sequential number designation as close as possible to the literature position citation. For example, "Jin et al. [XX]"

2. Fault Diagnoses Based on Improved 1DMCCNN Algorithm

The content of the chapter is relatively brief and requires interpretation by the reader. Please support the interpretation of the content by including an indication of the nature of the information, for example – methodology….

Line 91-106. There is a lack of reference in the text of the article to mathematical formulae 1 and 2.

Line 142. There is a lack of reference in the text of the article to mathematical formulae 5.

Line 132 The figure is clearly legible and complements the information in the chapter content well.

Line 150. The figure is clearly legible and complements the information in the chapter content well.

Line 146."....1DMCCNN-v1..." The designation „V1” should be explained (for example version 1?)

Line 154. „…1DMCCNN-v2….” The designation „V2” should be explained (for example version 2?)

Line 167.  The figure is clearly legible and complements the information in the chapter content well.

3. Simulations

In the chapter, detailed information about the test object appears: aircraft, landing gear, etc. This information was not articulated during the formulation of the research problem. This is a methodological error. Please take this into consideration.

Line 169. „LG hydraulic system” - please explain the abbreviation although it is common. Please explain the introduced abbreviations of terms, possibly as soon as they are introduced in the content of the article. A single explanation is sufficient in the execution of the explanation.

Line 169. „…LMS AMESIM….”- please explain the abbreviation although it is common. Please explain the introduced abbreviations of terms, possibly as soon as they are introduced in the content of the article. A single explanation is sufficient in the execution of the explanation.

Line 178. The figure is clearly legible and complements the information in the chapter content well.

Line 191. „…supply system 1 (Figure 2) and actuator 9 (Figure 2) were…” Please verify that the information actually refers to figure 2.

Line 194 – 195. Table 1. „/psi”, „/L”, „r/min”. The notation of the unit of measurement with a slash is colloquial and incomprehensible, for example "/psi". Please explain clearly the intention of this coding.

Indication of the unit of measure "r/min” using a fractional dash is acceptable but is colloquial in meaning. However, exponential notation should be used.

Please use the notation of the quotient in units of measure using mathematical notation with a power exponent for example: r·min-1

Line 206. Table 2. Indication of the unit of measure "L/min/bar” using a fractional dash is acceptable but is colloquial in meaning. However, exponential notation should be used.

Please use the notation of the quotient in units of measure using mathematical notation with a power exponent for example: L·min-1 ·bar-1

Line 212, 213. For example „0–2 s,” The unit of measure should be present with both range values.

Line 214, 215 "...Figure 5-1...." and similar. Such a figure number does not exist. Please change the way in which you inform the reader about the contents of the figure.

Line 220. The figure is clearly legible and complements the information in the chapter content well.

Line 225. The figure is clearly legible and complements the information in the chapter content well.

4.Analyses and Comparison 

Line 256. Table 5. „…lr ×= 0.9/epoch…” The notation with a slash is colloquial and incomprehensible. Please explain clearly the intention of this coding.

Line 275, 278, 280, Figure 8, 285, etc.. „…BP…” please explain the abbreviation although it is common. Please explain the introduced abbreviations of terms, possibly as soon as they are introduced in the content of the article. A single explanation is sufficient in the execution of the explanation.

Line 288. Figure 6. „size/MB” The notation with a slash is colloquial and incomprehensible. Please explain clearly the intention of this coding.

The content of the chapter achieves the veiled goal of the project well even though this goal has not been presented in detail in the preceding chapters.

5. Conclusion 

The content of the chapter in terms of conclusions should include those result elements that support the conclusions. Recalling these data will result in better indexing of the conclusions in relation to the research findings. Authors have previously revealed aspects of their research plans to the reader in an abrupt manner. Finally, it is worth concluding the article with clear indicators that confirm the success of the research study.
